# High-throughput micro-scale bandgap mapping for perovskite-inspired materials with complex composition space

Fang Sheng [1] ✉, Kangyu Ji [1,2] ✉, Linjie Dai [3,4], Alexander E. Siemenn [1], Eunice Aissi [1], Hamide Kavak[1,5], Basita Das[1], Tianran Liu[1,2], Shijing Sun [6] & Tonio Buonassisi[1]

To realize the full promise of high-throughput experimental workflows, the rate of sample synthesis must be matched by that of characterization. Of growing interest are contactless optical techniques that can rapidly measure material homogeneity and properties. Here, we present a hyperspectral imaging method to measure local optical bandgap distributions within samples, utilizing spatially-resolved reflectance spectra coupled with automated data analysis. We collect approximately one million optical bandgap data across the compositional space of $Cs_3(Bi_xSb_{1-x})_2(Br_yI_{1-y})_9$ perovskite-inspired materials. Our results show non-monotonic bandgap variations (i.e., bandgap bowing) along six composition gradient sequences, in addition to identifying samples with multiple bandgaps in statistics. High-throughput transient absorption spectroscopy reveals that within these compositions, the depletion of the ground state carriers to excited states occurred at discrete energy levels with independent carrier dynamics, consistent with the bandgap observation and indicative of phase separation. This work demonstrates the potential for rapid optical measurements to assess material quality and homogeneity in a high-throughput experimental setting, supporting screening and recipe optimization of optoelectronic material candidates with desired carrier dynamics and optical properties.

Material researchers are increasingly exploring automation hardware to improve synthesis throughput and reproducibility. Prominent examples include liquid-handling robots[1], microfluidic platforms[2,3], and combinatorial printers[4]. The increasing rate of sample fabrication must be matched by the rate of characterization, lest the latter represent a bottleneck to the overall learning rate. Traditionally, characterization methods are used to assess sample properties (e.g., ultraviolet-visible spectroscopy, UV-Vis) and to perform quality control (e.g., scanning electron microscopy, SEM, and X-ray diffraction,

XRD). Because traditional characterization hardware was developed for manual workflows, their throughputs are typically far below those of automated sample-synthesis hardware[5]. This motivates the search for higher-throughput characterization approaches that can perform property measurements and quality control, including sample homogeneity. In recent years, great effort has been made to develop high-throughput characterization capacities[5], focusing on both faster data acquisition and data analysis (Supplementary Fig. 1 and Supplementary Table 1). For example, time-evolved photoluminescence (PL)

[1]Department of Mechanical Engineering, Massachusetts Institute of Technology, Cambridge, MA, USA. [2]Research Laboratory of Electronics, Massachusetts Institute of Technology, Cambridge, MA, USA. [3]Department of Chemistry, Massachusetts Institute of Technology, Cambridge, MA, USA. [4]Cavendish Laboratory, University of Cambridge, Cambridge, United Kingdom. [5]Department of Physics, Cukurova University, Adana, Turkey. [6]Department of Mechanical Engineering, University of Washington, Seattle, WA, USA. ✉e-mail: shengf1227@gmail.com; axvcb1597382@gmail.com

characterization and automated peak-fitting algorithms have been developed to study the broadband emission in mixed Sn-Pb 2D perovskite microcrystals synthesized by pipetting robots[6]. Additionally, batch-processing algorithms[7] and machine learning[8,9] have been employed in data analysis pipelines, providing automated analysis of peaks in grazing-incidence wide-angle X-ray scattering (GIWAXS) and assisting identification of structural dimensions. However, a gap remains between the throughput of synthesis and characterization, posing a challenge for researchers to develop rapid characterization tools that can provide more accurate and comprehensive material properties.

A materials space that has been the subject of research interest and high-throughput exploration is halide perovskite-inspired semiconductors. These materials, which have a vast composition space[10,11], tend to form stoichiometric and defect-tolerant compounds, making them ideal for high-throughput study. In particular, the $A_3B_2X_9$ family, where $A$ is a monovalent cation (e.g., $Cs^+$), $B$ is a trivalent metal cation (e.g., $Bi^{3+}$ or $Sb^{3+}$), and $X$ is a halide anion (e.g., $Cl^-$, $Br^-$, or $I^-$), are explored as potential non-toxic materials for light harvesting[12]. However, several mixed-cation and mixed-anion alloys in this materials space have been shown to undergo nanoscale phase separation[13,14], affecting their properties and stability in optoelectronic devices[15,16].

To explore this materials space, we utilize a high-throughput combinatorial printer capable of printing 80-droplet samples of varying compositions in a gradient sequence within 2 mins, ~40 times faster than spin-coating. To extract the bandgap, we build upon an automatic bandgap extraction algorithm originally developed in prior work[17] (Fig. 1) and modify it to adapt to a more complex material space (Supplementary Fig. 2 and Supplementary Table 2). The data acquisition process follows the method used in prior work. Diffusive reflectance spectra are collected using a hyperspectral imaging setup, with a throughput of 80 samples every 5 mins. For data analysis, we develop a spatially resolved optical analysis technique to extract approximately 250 local bandgaps (at $140 \times 140\,\mu m$ resolution in this study; variable

depending on camera settings) for a single droplet sample. This technique absorbs maximum information from the scan and creates a statistical distribution of bandgaps for a given material composition, allowing us to evaluate intra-sample homogeneity. This accelerates optical characterization by a factor of over 6000 compared to conventional UV-Vis methods, which typically collect one bulk spectrum of a sample in 5 mins, and uniquely enables quantification of intra-sample variance. With our platform, we successfully explored the optical properties of perovskite-inspired materials in the quinary $Cs_3(Bi_xSb_{1-x})_2(Br_yI_{1-y})_9$ composition space, identifying composition ranges with high probabilities of phase separation.

## Results

Perovskite-inspired materials diverge from classic hybrid halide perovskite in how their optical properties are affected by 0D to 2D phase transitions[18] and chemical bonding[19], which can induce nonlinear bandgap changes in quasi-binary mixtures. Besides, the $A_3B_2X_9$ family has demonstrated outstanding humidity and thermal stability due to their vacancy-ordered structure compared to classic mixed halide perovskite[20,21]. It shows no degradation after the optical measurement (Supplementary Fig. 22) and can remain stable for months in the ambient storage environment. Given that they are under-studied and exhibit high stability, it is of interest to develop a platform capable of rapidly surveying the optical properties of composition ranges. We used a combinatorial printer to synthesize six quasi-binary composition-gradient sequences across the quinary material space $Cs_3(Bi_xSb_{1-x})_2(Br_yI_{1-y})_9$ (Fig. 1). To evaluate the reproducibility of gradient printing, we printed droplet arrays of fixed compositions, for 9 pure and mixed-composition compounds. We assessed the consistency of bandgaps through high-throughput optical characterization (Fig. 1a). High consistency in optical bandgaps was achieved for each batch-printed sequence (Supplementary Figs. 3 and 4). After quantifying reproducibility of fixed compositions, we then tested the reproducibility of gradient printing. We printed composition gradients in both forward and backward directions,

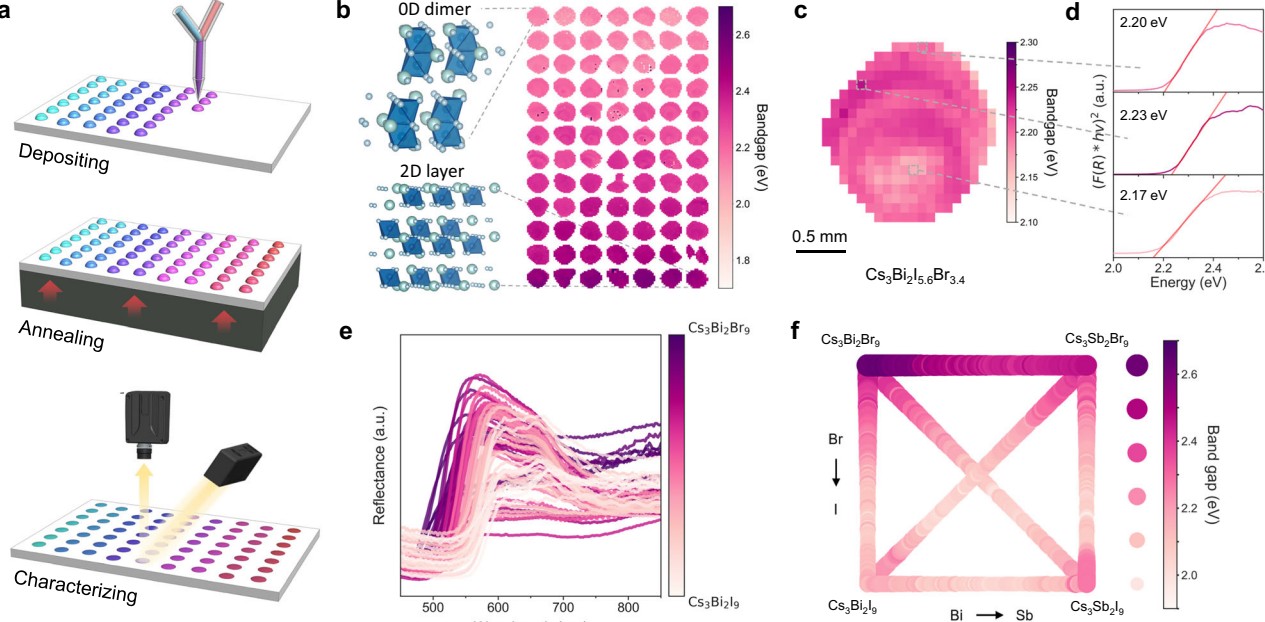

**Fig. 1 | High-throughput material composition exploration with combinatorial printing. a** Schematic of the high-throughput workflow has three stages: depositing the material with a linear composition gradient on the substrate, annealing the substrate to facilitate crystallization, and characterizing the optical properties. **b** High-throughput optical characterization of gradient-printed samples from $Cs_3Bi_2I_9$ (upper left, 0D dimer structure) to $Cs_3Bi_2Br_9$ (bottom right, 2D layered

structure). **c** Spatially resolved bandgap maps of one droplet sample. **d** The corresponding Tauc plots and fitting results of individual pixels in **c**. "a.u." denotes "arbitrary units". **e** Extracted reflectance spectra from the sequence $(Cs_3Bi_2Br_9)_x(Cs_3Bi_2I_9)_{1-x}$. "a.u." denotes "arbitrary units". **f** Bandgap extraction of six halide perovskite composition gradients through hyperspectral optical imaging.

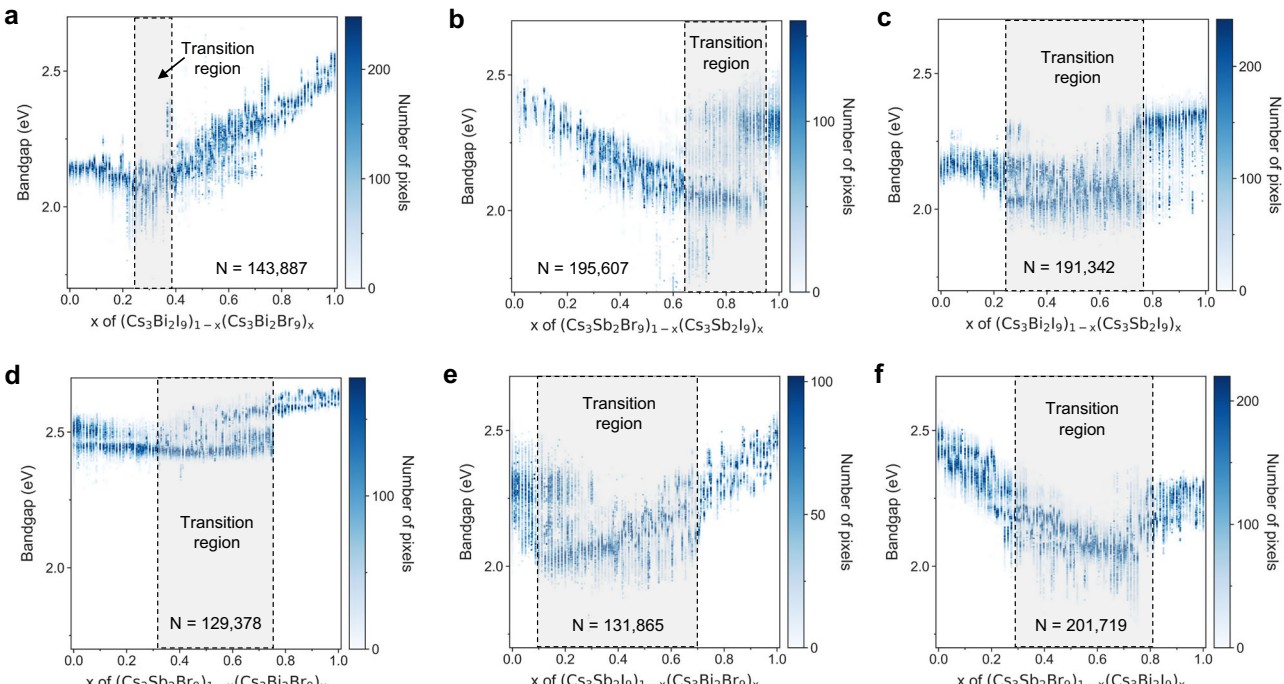

**Fig. 2 | Bandgap distribution of six gradient printing sequences of perovskite-inspired materials.** Compositions that lead to mixed phases are labeled in gray zones. $N$ is the number of data points collected from hyperspectral imaging for one sequence. **a** Sequence $(Cs_3Br_2I_9)_{1-x}(Cs_3Bi_2Br_9)_x$. **b** Sequence $(Cs_3Sb_2Br_9)_{1-x}$ $(Cs_3Sb_2I_9)_x$. **c** Sequence $(Cs_3Bi_2I_9)_{1-x}(Cs_3Sb_2I_9)_x$. **d** Sequence $(Cs_3Sb_2Br_9)_{1-x}$ $(Cs_3Bi_2Br_9)_x$. **e** Sequence $(Cs_3Sb_2I_9)_{1-x}(Cs_3Bi_2Br_9)_x$. **f** Sequence $(Cs_3Sb_2Br_9)_{1-x}$ $(Cs_3Bi_2I_9)_x$.

quantifying batch-to-batch bandgap coefficient of variance to be between 0.43% and 69.03% (Supplementary Figs. 5 and 6).

To determine the structural and elemental properties, we performed XRD and energy-dispersive spectroscopy (EDS) on the droplet samples (Supplementary Figs. 7 and 8). Based on prior studies[8], $Cs_3Bi_2Br_9$ and $Cs_3Sb_2Br_9$ adopt a hexagonal structure with $P\bar{3}m1$ space group, whereas $Cs_3Bi_2I_9$ and $Cs_3Sb_2I_9$ share a hexagonal structure but with $P6_3/mmc$ space group. In the Br compound, octahedral $[PbBr_6]^{4-}$ units form a 2D layered structure; in the I compound, two $[PbI_6]^{4-}$ exist as separate dimers and form a 0D dimer structure (Fig. 1b). Note that these two are difficult to distinguish by XRD, especially in samples with preferred orientation. Our XRD results reveal a gradual peak shift from $26.05°$ to $27.13°$ in the $Cs_3Sb_2I_9$ to $Cs_3Bi_2Br_9$ sequence, which we believe corresponds to the (006) plane of 0D $Cs_3Sb_2I_9$ and the (003) plane of 2D $Cs_3Bi_2Br_9$. The shift is attributed to the shrinkage of the perovskite lattice due to the reduced size of the halide ion from I to Br. An increase in crystallite sizes from $Cs_3Sb_2I_9$ to $Cs_3Bi_2Br_9$ is estimated based on the calculation of the Scherrer equation[22]. The EDS-inferred composition for each droplet confirms that the percentage of I elements decreases with a linear trendline along the gradient (Supplementary Fig. 8). In contrast to semiconductor transport properties (e.g., carrier lifetime, mobility) that depend strongly on defect structure, the optical bandgap (our target parameter of merit) is less sensitive to defects and remains nearly unchanged between spin-coated films and high-throughput printed droplets (Supplementary Fig. 21). Therefore, the bandgap trends observed in the $Cs_3(Bi_xSb_{1-x})_2(Br_yI_{1-y})_9$ material space using the high-throughput workflow are transferable to other synthesis methods.

We studied 6 composition gradients across the quinary composition space $Cs_3(Bi_xSb_{1-x})_2(Br_yI_{1-y})_9$ via high-throughput hyperspectral reflectance imaging. For each composition gradient, approximately 80 discrete droplets were printed on a glass slide. Every droplet was segmented by a computer-vision algorithm and assigned the corresponding precursor elemental ratio (i.e., composition; see Methods).

Tauc plots of droplets were calculated from reflectance data, and the optical bandgaps were then extracted using the reported method[23] (Fig. 1b, e). The results of 6 gradient-printing sequences are summarized in Fig. 1, with less than 0.1 eV standard error for each composition after three rounds of experiments (Supplementary Fig. 12).

If we average all pixels within a droplet, we implicitly assume the droplets are uniform, homogeneous, and phase-pure. However, we notice the printed droplets as well as the spin-coated films are not always spatially uniform due to crystallization kinetics and thermodynamic stability (Supplementary Note. 6 and Supplementary Fig. 11). By acquiring the spatially-resolved reflectance spectra, we are able to calculate the local bandgaps of small subregions ($140 \times 140 \mu m$) for each mm-sized printed droplet (Fig. 1c, d). Thus, bandgap statistics of local regions for the entire chemical gradient sequence can be extracted (Supplementary Fig. 13). We found droplets with high uniformity exhibit a bandgap range narrower than 0.08 eV, while those with lower uniformity exhibit ranges up to 0.30 eV (Supplementary Fig. 14).

The bandgap distributions within each droplet of a gradient sequence are plotted as a function of composition (Fig. 2). We collected 993,798 bandgap data points in total, with over 200 subregions per composition and around 480 droplets per gradient sequence. This vast dataset fully utilizes the massive volume of data generated from the spatially-resolved hyperspectral imaging technique, providing detailed insight into the bandgap behaviors within the composition space. The spatially-resolved bandgap statistics in Fig. 2 indicate that bandgap bowing[24], a phenomenon where the bandgap varies non-linearly with composition, occurs across all gradient sequences. This is particularly pronounced in sequences where the *B*-site remains constant and the *X*-site varies (Fig. 2a, b). The reason is the structural transition from 2D to 0D as the I concentration increases[18], where the reduced structural symmetry results in a higher optical bandgap. As shown in Fig. 2a, the bandgap initially decreases slightly and then increases as the Br percentage increases. This abnormal decreasing

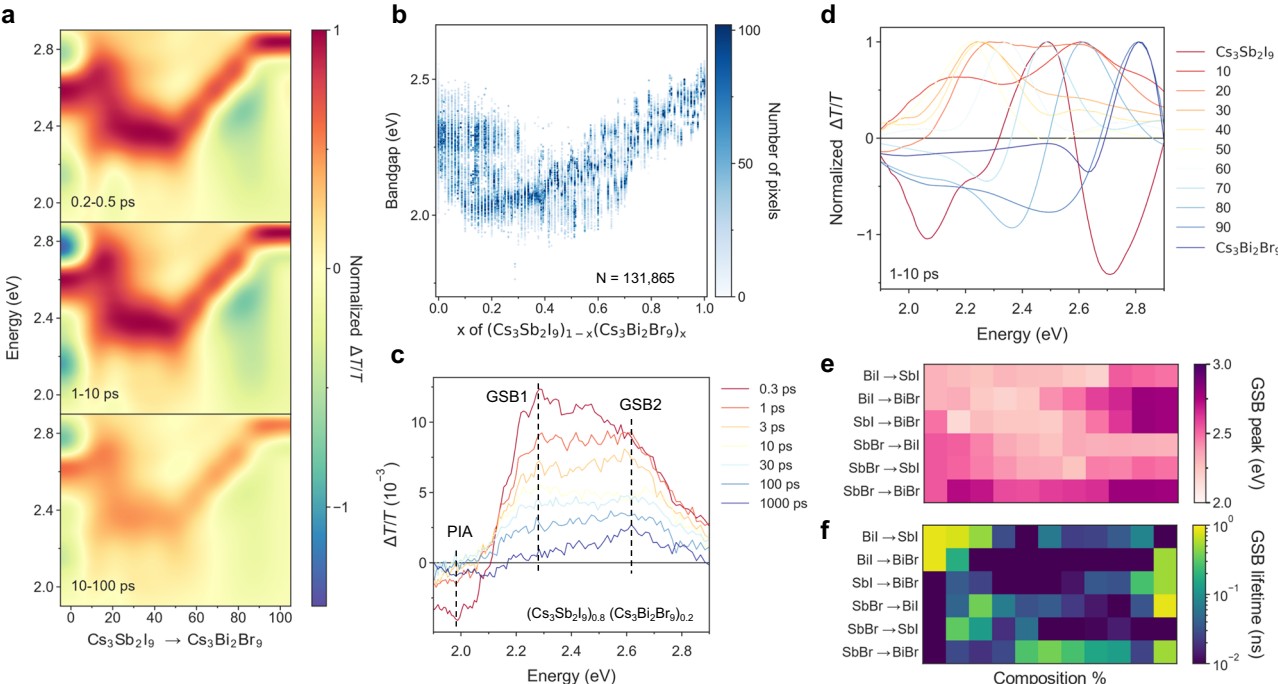

**Fig. 3 | Transient absorption spectroscopy on spin-coated films. a** 2D TA plot of spin-coated sequence $(Cs_3Sb_2I_9)_{1-x}(Cs_3Bi_2Br_9)_x$ at three different time scales. **b** Bandgap distribution of printed sequence $(Cs_3Sb_2I_9)_{1-x}(Cs_3Bi_2Br_9)_x$. **c** Integrated TA spectra (1–10 ps) of films with different compositions in $(Cs_3Sb_2I_9)_{1-x}$ $(Cs_3Bi_2Br_9)_x$. **d** TA spectra of $(Cs_3Sb_2I_9)_{0.8}$ $(Cs_3Bi_2Br_9)_{0.2}$ film at various time delays. **e** GSB peak positions of all spin-coated compositions in six sequences. **f** GSB lifetimes of all spin-coated compositions in six sequences. GSB: ground state bleach.

trend can be attributed to the improved structural symmetry during the 0D-to-2D transition, as higher dimensionality has been observed to lead to a lower bandgap in this material system[25]. Conversely, the bowing effect is less pronounced in sequences with fixed X-site and varying B-site (Fig. 2c, d). In these cases, the bowing stems from the energy disparity between the s and p orbitals of Sb and Bi atoms[19,26]. The valence band is dominated by Sb-5p orbitals and p orbitals of halides (Br-5p orbitals or I-6p orbitals), but the conduction band is mainly affected by Bi-6p orbitals and p orbitals of halides (Br-5p orbitals or I-6p orbitals). As the Sb ratio increases, the valence band and conduction band of the material drop at divergent rates, leading to nonlinear changes in the bandgaps. When both B-site and halide elements are varied (Fig. 2e, f), bandgap bowing is driven by both structural and chemical changes.

By examining variations in optical data within each drop-casted sample (i.e., the vertical variance in Fig. 2 at each composition), one can identify "transition regions" with multiple optical bandgaps coexisting and/or a relatively broader distribution of bandgaps (Fig. 2, gray zones). The observation of multiple optical bandgaps within certain compositional regions is highly suggestive of more than one semiconductor phase co-existing with a single printed droplet. Unfortunately, the small volumes of material make it challenging to perform powder XRD for unambiguous phase identification (see Supplementary Figs. 9 and 10 and Supplementary Note 3). The observation of broadening of a single bandgap is consistent with the possibility of phase segregation.

To advance mechanistic interpretations of the observed variance, we draw upon prior studies performed on similar compositional ranges using lower-throughput experiments and simulations. In the mixed-halide sequence $(Cs_3Sb_2I_9)_{1-x}(Cs_3Bi_2Br_9)_x$, the spatially resolved bandgap values exhibit this effect between $x = 0.64$ and $0.94$; two bandgap peaks are observed at 2.10 and 2.35 eV (Fig. 2b). These values are closely aligned with the bandgaps of 2D $Cs_3Sb_2I_9$ (2.05 eV) and 0D $Cs_3Sb_2I_9$ (2.36 eV)[27], suggesting a 0D-2D transition region. Similar 0D-

2D transition regions were observed in other sequences (Fig. 2a, e, f). The transition region of $(Cs_3Bi_2I_9)_{1-x}(Cs_3Bi_2Br_9)_x$ at $x = 0.25–0.37$ (Fig. 2a) is consistent with literature results on spin-coated and vapor-deposited films[28,29]. For $(Cs_3Sb_2Br_9)_{1-x}(Cs_3Bi_2I_9)_x$, the abrupt increase in the 2D phase occurs at $x = 0.7–0.8$ (Fig. 2f), consistent with the thin-film samples[8]. In $(Cs_3Sb_2I_9)_{1-x}(Cs_3Bi_2Br_9)_x$ and $(Cs_3Sb_2Br_9)_{1-x}(Cs_3Bi_2I_9)_x$, the mixed-phase regions include $x = 0.3–0.7$ range apart from the compositional range with discontinuous jump in bandgaps. The same phenomenon was observed in composition gradients with constant halides, where the bandgap distribution diverged into two branches at $x = 0.23–0.78$ and $x = 0.26–0.74$, respectively (Fig. 2c, d). A potential reason is B-site metal segregation or X-site halide segregation. When two compositions are mixed at comparable amounts, there will be the existence of distinct Bi-rich and Sb-rich domains or Br-rich and I-rich domains[30].

To further substantiate our findings, we performed broadband transient absorption spectroscopy (TA)[31] on thin-film samples of select compositions of interest (Figs. 3 and 4). With 10% compositional increment for each sequence, a total of 58 spin-coated films were synthesized and measured. The optical bandgaps of spin-coated films are highly aligned with printed droplets. TA spectroscopy provides insight into the carrier dynamics of a material by monitoring the change of absorption spectrum after 400-nm excitation. Figure 3a shows the TA results of the sequence $(Cs_3Sb_2I_9)_{1-x}(Cs_3Bi_2Br_9)_x$ at three different time ranges: 0.2–0.5 ps, 1–10 ps, and 10–100 ps. The red area signifies the ground state bleach (GSB), indicating the bleached states available from the ground state within the materials. We observed that the lowest-energy GSB is consistent with the steady-state optical bandgap from the hyperspectral measurements (Fig. 3a, b). GSB peaks at 2.28 and 2.62 eV were detected at composition $x = 0.2$ (Fig. 3c), consistent with the two bandgaps observed at 2.06 and 2.32 eV (Fig. 3b). Each GSB peak exhibits distinct carrier dynamics: the GSB peak at the lower energy dissipates more rapidly than the one at higher energy (Fig. 3d, see decay curves in Supplementary Fig. 20), suggesting

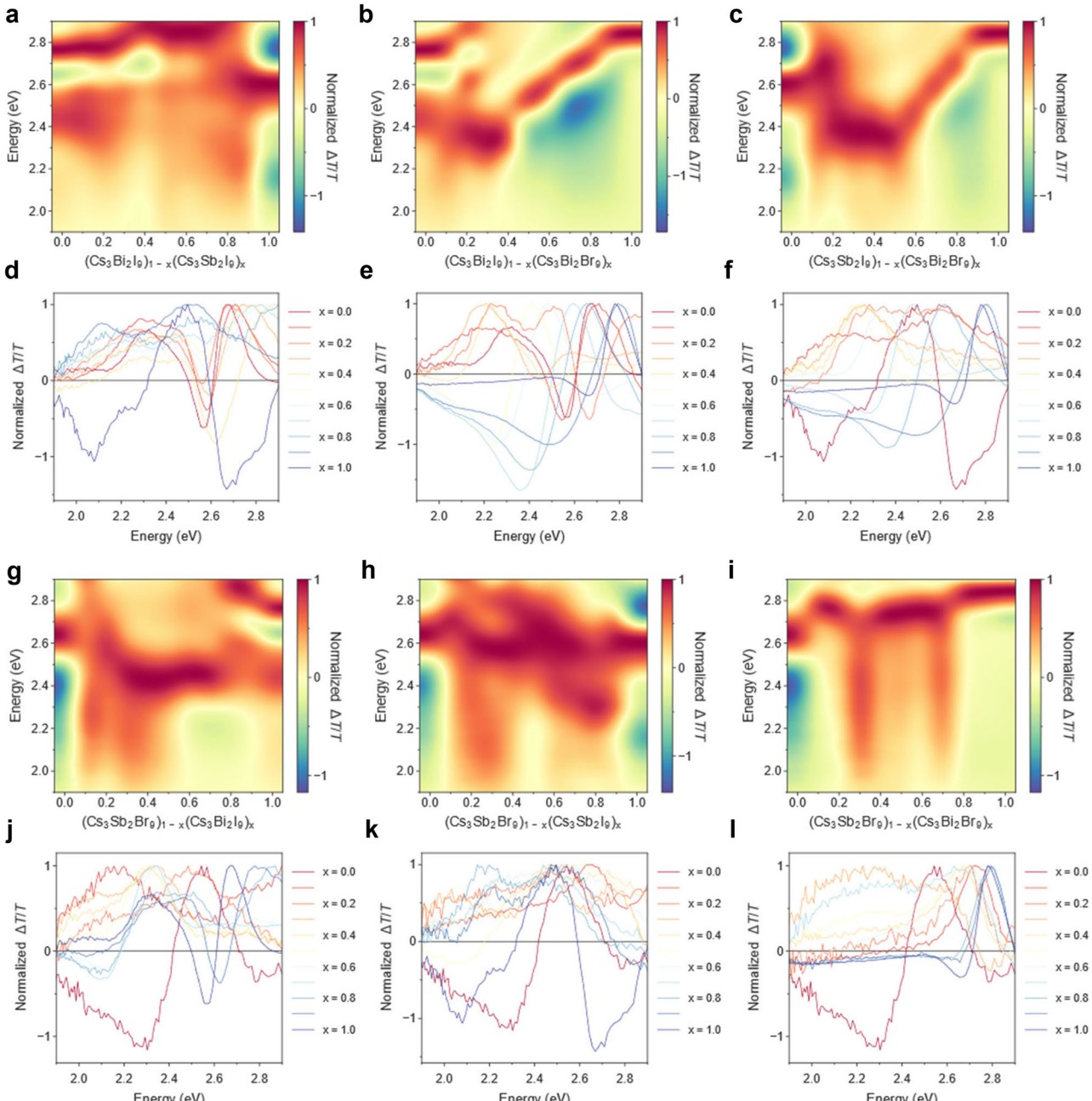

**Fig. 4 | 2D plots for high-throughput TA spectroscopy on spin-coated films from six gradient sequences of perovskite-inspired materials.** The TA signal was collected as fast as possible (5 mins per sample) without scan repetitions and data averaging. In total, 58 TA spectra were obtained with a 10% compositional increment in each sequence. Bicubic interpolation smoothing was performed to obtain the TA mapping results in **a–c** and **g–i**. The normalized changes in transmission at 1–10 ps were plotted against photon energy for **d–f** and **j–l**.

an absence of energy transfer from the higher to the lower energy state. This supports our findings that both 2D and 0D structures co-exist in the mixed compositions, as the carriers in the 0D structure are typically localized and do not tend to undergo charge transfer[32].

We then calculate the lowest GSB peak positions with the corresponding lifetimes for all compositions (Fig. 3e, f). We found that Bi-based perovskites exhibit longer carrier lifetimes than Sb-based ones, and Br-based perovskites typically have longer lifetimes than I-based ones. Moreover, the carrier lifetime in the bandgap bowing region is typically lower than that of the two parent materials, consistent with a higher density of defect states within the grain boundaries in the

intermediate composition region. Given the similar spectral characteristics across three rounds of sweeps and the absence of degradation during measurement (Supplementary Fig. 18), TA spectroscopy was conducted in a high-throughput manner, with each film scanned only once and an average measuring time of approximately 10 min. TA spectra in the time range of 1–10 ps were summarized in Fig. 4. Notably, the changing trends of GSB align consistently with the bandgap trends illustrated in Fig. 2. Besides, the compositional range with the presence of two distinct GSB areas and the broadening of GSB areas correlates closely with the mixed-phase regions observed in the bandgap distribution.

## Discussion

We have demonstrated an automated characterization platform that leverages combinatorial printing and high-throughput hyperspectral imaging. Our platform is able to measure and extract more than 100,000 bandgaps within an hour on a laboratory laptop, reducing the gap between synthesis and characterization rates. Through automated statistical analysis, we unveiled the complex optical properties in the quinary composition space of $Cs_3(Bi_xSb_{1-x})_2(Br_yI_{1-y})_9$ and defined the composition regions where phase segregation likely occurs.

Our approach has limitations. In gradient printing, the composition of droplets does not adhere to a linear gradient due to the complexities of fluid delivery, leading to errors in the composition estimation. In addition, our exploration was confined to only 6 linear gradient sequences, and the sampling bias will lead to difficulty in downstream tasks that use our data for property prediction of the entire composition space. While our work provides valuable optical analysis into the complex compositional space, achieving precise control over phase purity remains challenging, especially where halide segregation occurs in mixed halide perovskite compositions[15]. Bulk structural characterization methods such as XRD are complicated when the lattice structures are similar and the subsequent peak-fitting process is subjective (Supplementary Note 3). As fabricating phase-pure materials in high-throughput synthesis is usually difficult, developing an all-in-one testing platform that can perform structural-chemical-elemental characterization in a high-throughput manner is critical to accelerate material discovery.

Our work in high-throughput optical characterization opens a way to quantify the effective compositions in mixed-phase materials, enabling rapid screening of materials aimed at good carrier dynamics and optical properties. We believe this will facilitate rational material screening and synthesis recipe optimization for emerging optoelectronic applications.

## Methods

### Materials

N, N-Dimethylformamide (DMF, anhydrous, 99.8%), dimethyl sulfoxide (DMSO, anhydrous, 99.9%), isopropanol (IPA, anhydrous, 99.9%), and cesium iodide (CsI, 99.9%) were purchased from Sigma-Aldrich. Cesium bromide (CsBr, 99.9%) was purchased from Alfa Aesar. Bismuth bromide ($BiBr_3$, 99.999%), bismuth iodide ($BiI_3$, 99.999%), antimony bromide ($SbBr_3$, 99.995%), and antimony iodide ($SbI_3$, 99.999%) were purchased from Thermo Scientific. All chemicals were used without further purification.

### Film deposition

The substrates were cleaned with deionized water, 2% Hellmanex III solutions, and then IPA for 5 min under sonication. After drying, the substrates were transferred to an Ossila UV-ozone cleaner (L2002A3) for a 5-min treatment. The $Cs_3Bi_2I_9$, $Cs_3Sb_2I_9$, $Cs_3Bi_2Br_9$, and $Cs_3Sb_2Br_9$ precursor solutions were prepared by dissolving CsI/CsBr and $BiI_3$/$BiBr_3$/$SbI_3$/$SbBr_3$ in DMF/DMSO (4:1 v/v) with the stoichiometry of 3:2. All precursor solutions containing $Cs_3Sb_2Br_9$ were prepared at a concentration of 0.2 M, while the remaining precursor solutions were prepared at a concentration of 0.3 M. For each sequence gradient, 10% increment in compositions was taken for spin coating (i.e., $A_{1-x}B_x$, $x = 0$, 0.1,...,1). The final perovskite solution was prepared by mixing perovskite precursors with the corresponding stoichiometry ratio. The solution was spin-coated onto the glass substrates at 4000 rpm for 30 s. The films were annealed at 100 °C for 10 min in air and then transferred in a desiccator for storage.

### High-throughput combinatorial printing

0.1 M $Cs_3Bi_2I_9$, $Cs_3Sb_2I_9$, $Cs_3Bi_2Br_9$, and $Cs_3Sb_2Br_9$ precursor solutions in DMF were fed into the home-built combinatorial printer[26] to synthesize 6 gradient-printed sequences and 9 batch-printed sequences.

The plumbing lines of the printer were thoroughly cleaned with DMF solvent before each printing. For gradient printing, two of the precursors were purged at equal rates from both syringes for 50 s to reach the same initial status. The precursors were then pumped out of the syringes at pre-programmed rates with a linear composition gradient and deposited onto the substrate. For batch printing, the precursor was pumped out at a constant rate. The print head moved at 38 mm/s across the $75 \times 50$ mm glass slide in a rastering motion, depositing around 80 distinct composition droplets. The estimated volume of precursor solution for depositing one droplet is 1 μL. The droplet samples were then annealed at 150 °C for 15 min and stored in a desiccator. Higher annealing temperatures and longer annealing times were used for the droplet samples due to their larger thickness. The different processing conditions compared to the film samples had minimal impact on the bandgap behavior and phase separation phenomenon.

### Hyperspectral reflectance imaging

The spatially-resolved reflectance images of perovskite samples were obtained using a benchtop hyperspectral camera (Resonon Pika L) coupled with a halogen light source (Techniquip 21DC PowerLine). All samples were measured in air without encapsulation. The spectral range of the hyperspectral camera was set at 400-1200 nm with a 1 nm step size. For automatic bandgap extraction, we used a modified approach from a prior study[17] (Supplementary Note 4). After segmenting the droplets from the background, the coordinates of each pixel were recorded and the corresponding spectrum was extracted. The raw spectrum was transferred into the Tauc plot based on the Kubelka-Munk equation[23]. The bandgap value was then determined by the intersection between the linear fitting of the steepest slope and the background.

### X-ray diffraction

XRD patterns of droplet samples were obtained using a Rigaku Smar-tLab X-ray diffractometer with Cu-Kα sources. The range was set at 10–60° 2θ angle with a step size of 0.02°. The reference XRD pattern was simulated from the crystallographic information files of the corresponding material obtained from Materials Project[33]. The crystallite size was calculated from the raw data with the open-source Profex software[34].

### Scanning electron microscope with energy-dispersive X-ray spectroscopy

The SEM-EDS data were collected on 83 spots in a printed sample using a Zeiss Gemini 360 FE-SEM equipped with an Oxford Instruments Ultim Max 100 EDS detector. Each measured point was collected from the center of the individual droplet using a 20 keV electron beam and contains 1,000,000 X-ray counts. The spectrum data were then quantified in Oxford Aztec software via comparison to the following certified microanalysis standards: CsBr (for Cesium and Bromine); Bi metal; and RbI (for Iodine).

### Transient absorption spectroscopy

The output of a Ti:sapphire amplifier system (Spectra Physics Solstice Ace) operating at 1 kHz and generating ~100-fs pulses was split into the pump and probe beam paths. The 400-nm pump pulses were created by sending the 800-nm fundamental beam of the Solstice Ace through a second harmonic generating (SHG) beta barium borate (BBO) crystal of 1-mm thickness (Eksma Optics). The pump was blocked by a chopper wheel rotating at 500 Hz. For femtosecond to nanosecond measurements, the ultraviolet-visible broadband beam (330–700 nm) was generated by focusing the 800-nm fundamental beam onto a $CaF_2$ crystal (Eksma Optics, 5 mm) connected to a digital motion controller (Mercury C-863 DC Motor Controller) after passing through a mechanical delay stage (Thorlabs DDS300-E/M). The transmitted pulses were collected

with a monochrome line scan camera (JAI SW-4000M-PMCL, spectrograph: Andor Shamrock SR-163) with collected data fed straight into the computer. All samples were measured in ambient conditions under a 3.1-eV pump with a fluence of ~200.6 $\mu J/cm^2$. During the measurement, each sample was scanned only once for 10 min (aside from the four pure "corner" compositions, which were scanned three times each to ascertain phase stability during the measurement).

## Data availability
Source data are provided with this paper. The characterization data generated in this study have been deposited in the OSF database under accession code https://osf.io/nx2ae/. Source data are provided with this paper.

## Code availability
The original Autocharacterization Python code is available at https://github.com/PV-Lab/Autocharacterization-Bandgap (ref. 17). We modified the bandgap fitting algorithm for better generalizability and the code available at https://github.com/PV-Lab/High-throughput-micro-scale-bandgap-mapping (ref. 35).

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

## Acknowledgements

This project was primarily funded by First Solar, Inc. A.E.S., E.A., and T.L. acknowledge Eni S.p.A. through the MIT Energy Initiative,

TotalEnergies, and U.S. Department of Energy's Office of Energy Efficiency and Renewable Energy (EERE) under the Solar Energy Technology Office (SETO) Award Number DE-EE0010503, respectively.

## Author contributions

S.S. and T.B. conceived this work. E.A. and A.E.S. provided the printing hardware and autocharacterization algorithm resources. F.S. fabricated the samples. F.S. and H.K. collected the hyperspectral reflectance data. K.J. developed the local bandgap analysis methodology. F.S. performed X-ray diffraction. L.D. and K.J. performed the transient absorption spectroscopy. F.S., K.J., and L.D. analyzed the results. B.D. and T.L. helped the discussion. K.J. and T.B. supervised the research activity. F.S. and K.J. wrote and all authors edited this manuscript. This project was partially funded by First Solar, a manufacturer of solar photovoltaic panels.

## Competing interests

The authors declare no competing interests.
