## [Transparent Peer Review file · Nature Communications]

High-throughput micro-scale bandgap mapping for perovskite-inspired materials with complex composition space

Corresponding Author: Ms Fang Sheng

Version 0:

Reviewer comments:

Reviewer #1

(Remarks to the Author)

The study by Sheng and Ji et al. presents a hyperspectral imaging method for rapid, contactless characterization of optical bandgap distributions in perovskite-inspired materials. Using $\text{Cs}_3(\text{Bi}_x\text{Sb}_{1-x})_2(\text{Br}_y\text{I}_{1-y})_9$ as the model system, the authors claimed that they collected approximately one million bandgap data points across six compositional gradients. The results revealed non-monotonic bandgap changes (bandgap bowing) and phase separation regions with multiple bandgaps. High-throughput transient absorption spectroscopy supported these findings by identifying distinct energy levels and carrier dynamics. This work highlights the potential of automated optical techniques for screening and optimizing optoelectronic materials by characterizing their homogeneity, quality, and desired carrier properties. The novelty or strength of the work is the integration of hyperspectral imaging with combinatorial printing which can significantly accelerate the material characterization process. Also, leveraging machine learning for data analysis and automated bandgap extraction is forward-looking and aligns with trends in materials research. Although the work is of interest to a broad community of automated and high throughput systems, it has several weaknesses as outlined below which makes it unacceptable for publication in Nature Communications.

- 1- Scalability of the methods in the manuscript is not sufficiently discussed. It would be great to know if the results from the droplets can be transferred to spin coating or industrial deposition methods.
- 2- The authors stated they found droplets with high uniformity exhibit a bandgap range narrower than 0.08 eV, while those with lower uniformity exhibit ranges up to 0.30 eV. (Supplementary Fig. 8). Could the author clarify if the thickness of the film can affect the small change on band gap. Usually, the droplets are thicker on the outer circle than the inner due to rate of solvent evaporation.
- 3- For bandgap bowing indicated in figure 2 a and b, the authors attributed the phenomena to the structural transition from 2D to 0D as the I concentration increases¹³, where the reduced structural symmetry results in a higher optical bandgap. However, the band gap trends in Figure 2 a) and b) are different. In 2a) as the Br increases, Eg increases as expected, in 2b) as the I increase, Eg decreases first and then increases.
- 4- Again, in lines 148 and 149, after describing figure 2b), the authors wrote: 2b). These values are closely aligned with the bandgaps of 0D $\text{Cs}_3\text{Sb}_2\text{I}_9$ (2.05 eV) and 2D $\text{Cs}_3\text{Sb}_2\text{I}_9$ (2.36 eV)²⁰, suggesting a 0D-2D transition region. There must be a typo here and the second composition should be $\text{Cs}_3\text{Sb}_2\text{Br}_9$ (Eg: 2.36 eV).
- 5- After discussing Figure 2 c) and d) (lines 157-160). Authors wrote: A potential reason is B-site metal segregation or X-site halide segregation. When two compositions are mixed at comparable amounts, there will be the existence of distinct Bi-rich and Sb-rich domains or Br-rich and I-rich domains. This is consistent with the presence of asymmetric diffraction peaks from power XRD measurements of the mixed halide sample. This explanation is insufficient here. Can you provide more information why asymmetric XRD peaks can be associated with B-site cation or halide rich domains?
- 6- For samples used for the TA spectroscopy, it is mentioned that samples are spin coated. It is unclear if all compositions are spin coated? How many samples are spin coated? How different are spin coated samples vs printed droplets?
- 7- Figure 4 caption how can you do gradient composition by spin coating? The caption explains: high-throughput TA spectroscopy on spin-coated films from six gradient sequences of perovskite inspired materials. It is unclear whether the samples are manually spin coated? How many samples are synthesized to make the 2D maps.

8- In line 193, the author stated that the TA was also conducted in a high throughput way, but it is not explained how.

9- There is inconsistency in the explanation of the number of samples which have been characterized. In the abstract says 1 million samples and in discussion says our platform is able to measure and extract more than 100,000 band gaps within 1 hr. Please clarify.

10- In the materials and methods part, high throughput combinatorial printing, what is the volume of droplets as compared to spin coated samples. There is also a 50 C difference in annealing temperature of droplets and spin coated samples. How is this affecting or not affecting on crystallinity or phase segregation?

11- The authors should also provide information on the stability of their synthesized compositions. How stable are they?

12- The authors should add more citations of recent works of other research groups focusing on high throughput synthesis of halide perovskites integrated with high throughput optical characterizations.

13- In supplementary figure 14, which data/figures is gradient printed, and which one is spin coated samples. please clarify that.

14- The authors claim that they have generated 100,000 band gap data, no ML analysis has been done on the data. I realized they have done automatic bandgap extraction. What is the reason for not using ML. I agree that we do not necessarily always need to perform ML algorithms, but it would be great to provide some thoughts on this.

(Remarks on code availability)

Reviewer #2

(Remarks to the Author)

Sheng and Ji et al present a concise study outlining the importance of developing high-throughput (HT) analysis techniques to complement high-throughput experimental capabilities. This work uses perovskite-inspired materials to explore a wide range of compositions through HT workflows and subsequently analyzes their bandgaps through a hyperspectral imaging method connected with automated analysis. The primary focus of the research itself is on the optoelectronic properties and how variations in bandgap are related to different material compositions along with the effects of phase separation. Overall I find the article is appropriately detailed, thorough and presents new insights for the range of compositions explored, specifically in terms of the effects of nanoscale phase-separation on bandgap. I believe in terms of content and execution it is appropriate for publication in Nature Communications pending some minor modifications and some additional discussion/foreword.

Though I believe it could go beyond the scope of a communication, some additional commentary on other high throughput methods would be appropriate to add, particularly to highlight the relative speed, throughout, and presumably non-destructive nature of the presented technique versus alternative experiments.

In addition, there is a great deal of emphasis on the number of data points collected and in what timeframe. This would be well complemented by discussion about how many data points are necessary. This discussion is partially present in the SI through the experiments exploring batch to batch consistency but could be added in a concluding sentence in the discussion or through analysis of subsets of the data collected to identify the necessary minimum for the system. Further scaling and extension to infinite parameter spaces or replicates is not necessarily desirable or practical and I believe more explicit commentary on how much data needs to be collected to be meaningful would be worthwhile.

By extension it would be insightful to have the authors more explicitly outline their selection criteria for pursuing specific samples with the slower techniques that are necessary for more complete characterization of a given composition.

Prior to publication I believe reference 26 must be updated. Presently it points to a pre-print published by several of the current authors that explores a different perovskite system using what I believe are the same droplet-based HTS system and the same hyperspectral autocharacterization system used in this work. The pre-print in question was published in Nature Communications earlier this year (Siemenn, A.E., Aissi, E., Sheng, F. et al. Using scalable computer vision to automate high-throughput semiconductor characterization. Nat Commun 15, 4654 (2024)). I believe this study extends that one and serves as an excellent follow-up demonstration of the system capabilities. As a follow-up I believe this study should highlight the pre-existing nature of the tools more prominently and should include some discussion of how the creation of datasets using pre-existing tools will increasingly become important for this type of research. Presently reference 26 is only featured in the methods and the SI at the end of the article. I believe it should be introduced at the beginning of the article in the introduction. The authors should also take the time to explain the novelty between this article and the current manuscript. What is the new contribution to knowledge that the current manuscript brings and how does that go beyond the article published earlier this year highlighting why they both deserve to be published in nature communications?

(Remarks on code availability)

Reviewer #3

(Remarks to the Author)

(Remarks on code availability)

Reviewer #4

(Remarks to the Author)

Sheng, Ji et al. describe a high-throughput setup consisting of a rapid inkjet-printing-like setup complemented by diffuse reflectance and further augmented by transient absorption spectroscopy on perovskite-inspired Cs/Bi/Sb/I/Br mixes.

Overall the study is interesting, but there remain several points that need clarification, before I can recommend publication.

1. Visualization: While the color-maps are aesthetic, in some cases they make it hard to properly read the graphs. Prime examples are Figure S1 and S2 (white on a white background ...) Histograms would be much more meaningful (for Fig. S2, I'd even suggest correlation curves comparing the different rounds, much simpler)
2. the observation of a high degree of heterogeneity within a single droplet is an interesting observation. The authors should provide more information here. Is the (in-)homogeneity driven by a particular parameter? precursor concentration, pre-agitation, annealing time/temperature?
For some select samples (e.g., most vs. least homogeneous) more in-depth characterization would be valuable to support the "general" validity of the HT results obtained.. e.g.,
3. the "mixed zone" bandgap regions observed are highly interesting. For many 3D ABX₃ perovskites phase separation is a dynamic process driven by light, heat, etc. Are the phases directly separated here? Considering the extremely fast characterization (at least the UV-Vis part) an observer-effect can be likely excluded...? E.g., by measuring the same sample before and after a short light ageing interesting information could be gained and also highlight another feature of the workflow, without me asking for this to be done for all samples, of course, but e.g. Fig. 2c provides a broad "mixed phase region".
4. For some of the select mixes with multiple bandgaps the actual spectra would be highly appreciated. As far as I know the work from the group, systems with multiple bandgaps haven't come up yet and hence I am a bit skeptical as to how the algorithm deals with that?
5. complementary XRD would be also very valuable to show the presence of two phases along the same lines of thinking.
6. I am not sure why the authors chose to perform TAS on the samples, as there is no real additional information derived from it in my opinion.
The authors write: "We observed a strong correlation between the lowest-energy GSB and the steady-state optical bandgap from the hyperspectral measurements" and I "visually" agree. Why did the authors not simply correlate the two?
7. The pump fluence of the TA measurement is very high (200 uJ/cm², roughly 10.000x sun equivalent), so there is a high chance of measurement-induced artifacts, can the authors exclude that, e.g., by sequential measurements (I read the "The transient absorption spectroscopy was also conducted in a high-throughput way. Each film was scanned only once and the average measuring time was around 10 min" paragraph, but I doubt it.
8. can the authors correlate what they write to be correlated? e.g., GSB lifetime with "disorder" in the bowing region?
9. Another problem is that all samples are being treated the same here, but depending on the bandgap, bandgap-type (direct/indirect), quantum confinement, etc. things change.
E.g., somewhat counterintuitively, at high fluences, for a direct-bandgap semiconductor without confinement (and it is not clear which of the compositions are which, so confined, direct or indirect bandgap?), one enters the radiative regime, which in turn leads to faster (compared to trap-assisted) recombination. Similarly confined vs. 3D systems behave vastly different in their ultrafast dynamics, so I am highly skeptical about their comparability in the way it is presented.

(Remarks on code availability)

not tested

Version 1:

Reviewer comments:

Reviewer #1

(Remarks to the Author)

The authors addressed my comments and suggestions. I believe the manuscript is ready for publication in Nature Communications.

(Remarks on code availability)

I didn't review the codes

Reviewer #2

(Remarks to the Author)

I am satisfied with the authors response. i recommend publication.

(Remarks on code availability)

Reviewer #3

(Remarks to the Author)

(Remarks on code availability)

Reviewer #4

(Remarks to the Author)

I thank the authors for carefully considering all my comments. They answered my and the other reviewers' comments in a convincing and professional way giving a lot of thought to the answers and clearly indicated where and how they modified the manuscript, where they saw fit. As for my comments I can say that I am satisfied with the modifications and think the manuscript adds significantly to the body of knowledge of $\text{Cs}_3(\text{Bi}_x\text{Sb}_{1-x})_2(\text{Br}_y\text{I}_{1-y})_9$ double-perovskite and analogously inspired materials and offers a useful tool for characterization. I am looking forward to seeing the setup being coupled to other characterization tools or aging setups (should allow for unprecedented high-throughput insights into rapid aging dynamics) to fully exploit its capabilities.

(Remarks on code availability)

REVIEWER COMMENTS

Reviewer #1 (Remarks to the Author):

The study by Sheng and Ji et al. presents a hyperspectral imaging method for rapid, contactless characterization of optical bandgap distributions in perovskite-inspired materials. Using $\text{Cs}_3(\text{Bi}_x\text{Sb}_{1-x})_2(\text{Br}_\gamma\text{I}_{1-\gamma})_9$ as the model system, the authors claimed that they collected approximately one million bandgap data points across six compositional gradients. The results revealed non-monotonic bandgap changes (bandgap bowing) and phase separation regions with multiple bandgaps. High-throughput transient absorption spectroscopy supported these findings by identifying distinct energy levels and carrier dynamics. This work highlights the potential of automated optical techniques for screening and optimizing optoelectronic materials by characterizing their homogeneity, quality, and desired carrier properties. The novelty or strength of the work is the integration of hyperspectral imaging with combinatorial printing which can significantly accelerate the material characterization process. Also, leveraging machine learning for data analysis and automated bandgap extraction is forward-looking and aligns with trends in materials research. Although the work is of interest to a broad community of automated and high throughput systems, it has several weaknesses as outlined below which makes it unacceptable for publication in Nature Communications.

We thank the reviewer for the thorough read and suggestions. We appreciate the reviewer for pointing out the novelty of our work and the impact to a broad community.

1. Scalability of the methods in the manuscript is not sufficiently discussed. It would be great to know if the results from the droplets can be transferred to spin coating or industrial deposition methods.

Material properties heavily depend on synthesis methods and conditions, but some are intrinsic to the material and remain unaffected by the synthesis process. In this manuscript, the target parameter of merit, the bandgap, exhibits high consistency between high-throughput synthesized droplets and spin-coated films, as shown in **Supplementary Fig. 21**. Therefore, we can trust the workflow and conclude that the high-throughput bandgap mapping results are transferable to thin films despite significant morphological differences.

Action:

- We have now added the following explanations to the results session at Line 112 -117: “In contrast to semiconductor transport properties (*e.g.*, carrier lifetime, mobility) that depend strongly on defect structure, the optical bandgap (our target parameter of merit) is less sensitive to defects and remains nearly unchanged between spin-coated films and high-throughput printed droplets (**Supplementary Fig. 21**). Therefore, the bandgap trends

observed in the $\text{Cs}_3(\text{Bi}_x\text{Sb}_{1-x})_2(\text{Br}_y\text{I}_{1-y})_9$ material space using the high-throughput workflow are transferable to other synthesis methods.”

2. The authors stated they found droplets with high uniformity exhibit a bandgap range narrower than 0.08 eV, while those with lower uniformity exhibit ranges up to 0.30 eV. (Supplementary Fig. 8). Could the author clarify if the thickness of the film can affect the small change on band gap. Usually, the droplets are thicker on the outer circle than the inner due to rate of solvent evaporation.

Thank you for raising this important point. Thickness itself does not affect the bandgap, as the bandgap is an intrinsic optical property of the material. However, thickness non-uniformity, resulting from the crystallization mechanism of droplets, reflects compositional and structural variations, which lead to broad bandgap distributions. Since the printing conditions for each droplet are identical, the unavoidable thickness non-uniformity can be considered a systematic error. As shown in **Supplementary Fig. 3**, all droplets in a batch-printed sample exhibited highly similar bandgap distributions, despite variations in thickness. Pure compositions, such as $\text{Cs}_3\text{Bi}_2\text{Br}_9$, had a narrower bandgap distribution, while mixed compositions, such as $\text{Cs}_3(\text{Bi}_{0.5}\text{Sb}_{0.5})_2(\text{Br}_{0.5}\text{I}_{0.5})_9$, exhibited broader bandgap distributions due to structural and compositional non-uniformity. Furthermore, **Supplementary Fig. 5** indicates that bandgap distributions remained similar across multiple experimental rounds, even though droplet thicknesses varied from one round to another.

Action:

- We have now modified the texts in **Supplementary Note. 1** at Line 57-58: For example, air bubbles generated in plumbing lines during printing could result in differences in droplet sizes, shapes, and thicknesses.

3. For bandgap bowing indicated in figure 2 a and b, the authors attributed the phenomena to the structural transition from 2D to 0D as the concentration increases, where the reduced structural symmetry results in a higher optical bandgap. However, the band gap trends in Figure 2 a) and b) are different. In 2a) as the Br increases, E_g increases as expected, in 2b) as the I increase, E_g decreases first and then increases.

Generally, E_g decreases linearly as the iodine ratio increases. However, the reduced symmetry caused by the structural transition from 2D to 0D leads to an abnormal increase in E_g as the composition approaches the iodine-rich side¹⁸. Consequently, in **Fig.2a**, E_g slightly increases when the ratio of I increases from 0.7 to 1 (i.e. x decreases from 0.3 to 0). A similar increase in E_g is observed in **Fig.2b**, where the iodine ratio increases from 0.64 to 0.95.

Action:

- We now add clarification at Line 152-155: “As shown in **Fig. 2a**, the bandgap initially decreases slightly and then increases as the Br percentage increases. This abnormal decreasing trend can be attributed to the improved structural symmetry during the 0D-to-2D transition, as higher dimensionality has been observed to lead to a lower bandgap in this material system²⁵.”

[25] Gao, P., Bin Mohd Yusoff, A. R. & Nazeeruddin, M. K. Dimensionality engineering of hybrid halide perovskite light absorbers. *Nat. Commun.* **9**, 5028 (2018).

4. Again, in lines 148 and 149, after describing figure 2b), the authors wrote: 2b). These values are closely aligned with the bandgaps of 0D Cs₃Sb₂I₉ (2.05 eV) and 2D Cs₃Sb₂I₉ (2.36 eV)²⁰, suggesting a 0D-2D transition region. There must be a typo here and the second composition should be Cs₃Sb₂Br₉ (Eg: 2.36 eV).

Thank you for your careful reading. We apologize for the typo— the bandgap values of 2D and 0D Cs₃Sb₂I₉ should be swapped.

Action:

- We now corrected the main text at Line 188-189: “These values are closely aligned with the bandgaps of 2D Cs₃Sb₂I₉ (2.05 eV) and 0D Cs₃Sb₂I₉ (2.36 eV)²⁰.”

5. After discussing Figure 2 c) and d) (lines 157-160). Authors wrote: A potential reason is B-site metal segregation or X-site halide segregation. When two compositions are mixed at comparable amounts, there will be the existence of distinct Bi-rich and Sb-rich domains or Br-rich and I-rich domains. This is consistent with the presence of asymmetric diffraction peaks from power XRD measurements of the mixed halide sample. This explanation is insufficient here. Can you provide more information why asymmetric XRD peaks can be associated with B-site cation or halide rich domains?

By comparing the PXRD pattern of Cs₃(Bi_{0.5}Sb_{0.5})₂(Br_{0.5}I_{0.5})₉ with the reference patterns of 2D Cs₃Bi₂Br₉ and 0D Cs₃Sb₂I₉ (**Supplementary Fig. 10**), we observed that all peaks matched well with the 2D Cs₃Bi₂Br₉ reference, with a slight peak shift to lower two-theta angles. This indicates that Cs₃(Bi_{0.5}Sb_{0.5})₂(Br_{0.5}I_{0.5})₉ has a purely 2D structure. Meanwhile, all peaks exhibited broadening and asymmetry, which can be attributed to either mixed 2D and 0D phases, elemental segregation, or both. Since the composition adopts a 2D structure, the peak broadening and asymmetry are most likely due to elemental segregation. Bi and I are larger than Sb and Br, so Bi-rich and I-rich domains have expanded lattice parameters, resulting in left-shifted peaks in the PXRD pattern. The overlap of closely spaced peaks generates the broadened and asymmetric profile.

Action:

- We have now modified the texts in **Supplementary Note 3** at Line 104-111: “By comparing the PXRD pattern with reference patterns, we observed that the peaks in the experimental pattern matched well with the 2D $\text{Cs}_3\text{Bi}_2\text{Br}_9$ reference, with a slight peak shift to lower two-theta angles, implying that the mixed composition $\text{Cs}_3(\text{Bi}_{0.5}\text{Sb}_{0.5})_2(\text{Br}_{0.5}\text{I}_{0.5})_9$ has a purely 2D phase (**Supplementary Figure 10**). Additionally, peak broadening and asymmetry were observed even in the powdered sample, which could result from either mixed 2D and 0D phases or elemental segregation. Given the 2D structure of $\text{Cs}_3(\text{Bi}_{0.5}\text{Sb}_{0.5})_2(\text{Br}_{0.5}\text{I}_{0.5})_9$, the peak broadening and asymmetry are most likely due to elemental segregation. This conclusion is further supported by the varying bandgap observed in optical measurements.”

6. For samples used for the TA spectroscopy, it is mentioned that samples are spin coated. It is unclear if all compositions are spin coated? How many samples are spin coated? How different are spin coated samples vs printed droplets?

We used a 10% increment across six sequences, producing a total of 58 spin-coated samples covering all compositions. Due to differences in synthesis and crystallization processes, spin-coated films and printed droplets vary in thickness, crystallite size, and orientation. However, they share the similar phases and bandgaps which are intrinsic to the materials as indicated by **Supplementary Figure 21**. Spin-coated samples were used for TA measurements because performing TA measurements on thick samples is challenging.

Action:

- We have now clarified in the main text at Line 202-204: “With 10% compositional increment for each sequence, a total of 58 spin-coated films were synthesized and measured. The optical bandgaps of spin-coated films are highly aligned with printed droplets.”

7. Figure 4 caption how can you do gradient composition by spin coating? The caption explains: high-throughput TA spectroscopy on spin-coated films from six gradient sequences of perovskite inspired materials. It is unclear whether the samples are manually spin coated? How many samples are synthesized to make the 2D maps.

During the spin-coating process, precursor solutions of the pure compositions $\text{Cs}_3\text{Bi}_2\text{I}_9$, $\text{Cs}_3\text{Sb}_2\text{I}_9$, $\text{Cs}_3\text{Bi}_2\text{Br}_9$, and $\text{Cs}_3\text{Sb}_2\text{Br}_9$ were prepared by dissolving CsI/CsBr and $\text{BiI}_3/\text{BiBr}_3/\text{SbI}_3/\text{SbBr}_3$ in DMF/DMSO (4:1 v/v) with a stoichiometry of 3:2. A 10% step increment was used for each sequence (i.e., A_{1-x}B_x , $x = 0, 0.1, \dots, 1$), and the pure precursors were mixed stoichiometrically. The mixed precursor solutions were manually spin-coated onto glass substrates at 4,000 rpm for 30 seconds. The films were then annealed at 100 °C for 10 minutes. A total of 58 samples were

prepared, considering the four pure compositions. Each 2D map contained 11 data points and was plotted using interpolation.

Action:

- We have added more details to the Methods section at Line 309-310: “For each sequence gradient, 10% increment in compositions was taken for spin coating (i.e. $A_{1-x}B_x$, $x = 0, 0.1, \dots, 1$).”

8. In line 193, the author stated that the TA was also conducted in a high throughput way, but it is not explained how.

Thank you for your careful reading. High-throughput TA refers to scanning one sweep on each sample for 10 minutes. Generally multiple sweeps are required during the measurement in order to quantify beam damage, which limits the throughput. As shown in **Supplementary Fig. 18**, we conducted 2–3 sequential sweeps on four samples with pure compositions. The data from the first, second, and third sweeps exhibit no significant variation, either in the absolute intensity of the spectral features or in the kinetic profiles, indicating that the samples were not degraded by the laser exposure during the measurement process. Therefore, one sweep is adequate due to negligible beam damage. The TA measurement on 58 samples can be conducted in a high-throughput way by performing only one sweep.

Action:

- We have added clarification in the Methods section at Line 369-371: “During the measurement, each sample was scanned only once for 10 minutes (aside from the four pure “corner” compositions, which were scanned three times each to ascertain phase stability during the measurement).”

9. There is inconsistency in the explanation of the number of samples which have been characterized. In the abstract says 1 million samples and in discussion says our platform is able to measure and extract more than 100,000 band gaps within 1 hr. Please clarify.

Thank you for raising this point. We understand that some expressions in the manuscript—such as *samples vs. bandgap data points* and *throughput vs. total number of collected data points*—may be confusing. In this work, nearly 1 million spatially resolved bandgap data points were collected across six experimental sequences, as shown in **Figure 2** ($N = 993,798$). While the total number of bandgap data points reflects the overall dataset size, the throughput of our characterization method exceeds 100,000 bandgap data points per hour.

Action:

- We have now added a **Supplementary Note 5**, clarifying the workflow throughput and the total number of extracted bandgap data points: “The total number of bandgap data points collected was nearly 1 million, spanning six experimental rounds with three forward and three backward printing sequences. Each round involved printing approximately 85 droplets on a glass substrate. Although droplet sizes varied, each droplet typically contained 200 to 300 pixels. As shown in **Fig. 2**, N represents the number of bandgap data points for each sequence, with a total of 993,798 data points collected across all six sequences. The throughput of our characterization method is approximately 100,000 bandgap data points per hour, with the process involving hyperspectral camera scanning and spatially-resolved spectra extraction. Assuming each droplet has 200 pixels and that both scanning and extraction take 5 minutes, 102,000 ($200 \times 85 \times 6$) bandgaps can be measured and extracted in one hour.”

10. In the materials and methods part, high throughput combinatorial printing, what is the volume of droplets as compared to spin coated samples. There is also a 50 C difference in annealing temperature of droplets and spin coated samples. How is this affecting or not affecting on crystallinity or phase segregation?

The volume of the droplets is approximately 1 μL . Ideally, the annealing temperature should be the same for comparison. The reason for using different annealing temperatures is that longer annealing times are required for droplets due to their larger thickness when annealed at 100 °C. If droplets were still annealed at 100 °C for 10 minutes, remaining solvent molecules would affect the hyperspectral bandgap measurement results. Therefore, we chose higher annealing temperature. As indicated by **Supplementary Fig. 15**, the average bandgaps of the droplet and film samples are well aligned. **Supplementary Fig. 11** also shows the heterogeneity of bandgaps in spin-coated films, revealing the occurrence of phase separation. Thus, the bandgap values and phase separation phenomenon are not sensitive to differences in crystallization conditions and form factors.

Action:

- We have added clarification in the methods section at Line 323-328: “The estimated volume of precursor solution for depositing one droplet is 1 μL . The droplet samples were then annealed at 150 °C for 15 minutes and stored in a desiccator. Higher annealing temperatures and longer annealing times were used for the droplet samples due to their larger thickness. The different processing conditions compared to the film samples had minimal impact on the bandgap behavior and phase separation phenomenon.”

11. The authors should also provide information on the stability of their synthesized compositions. How stable are they?

The synthesized perovskite-inspired compositions demonstrated high stability compared to traditional perovskites. All synthesis and measurements were performed in an ambient environment without any degradation. As shown in **Supplementary Fig. 22**, no obvious change in bandgap distributions was observed in the printed perovskite-inspired compositions after 10 minutes of hyperspectral line light illumination. All samples were stored in a desiccator under ambient conditions, and no color change was observed after one month of storage.

In general, the $A_3B_2X_9$ family has demonstrated outstanding humidity and thermal stability due to their vacancy-ordered structure. Photovoltaic devices based on $A_3B_2X_9$ materials are capable of maintaining their initial performance over the long term^{1,2,3}.

1. Li, J., Lv, Y., Han, H., Xu, J. & Yao, J. Two-Dimensional $Cs_3Sb_2I_{9-x}Cl_x$ Film with (201) Preferred Orientation for Efficient Perovskite Solar Cells. *Materials* 15, 2883 (2022).
2. Shil, S. K. et al. Crystalline all-inorganic lead-free $Cs_3Sb_2I_9$ perovskite microplates with ultra-fast photoconductive response and robust thermal stability. *Nano Res.* 14, 4116–4124 (2021).
3. Hoye, R. L. Z. et al. Methylammonium Bismuth Iodide as a Lead-Free, Stable Hybrid Organic–Inorganic Solar Absorber. *Chem. – Eur. J.* 22, 2605–2610 (2016).

Action:

- We have now added more description on the stability of the synthesized materials to the results session at Line 85-88: “Besides, the $A_3B_2X_9$ family has demonstrated outstanding humidity and thermal stability due to their vacancy-ordered structure compared to classic mixed halide perovskite^{20,21}. It shows no degradation after 10 minutes of hyperspectral line light illumination (**Supplementary Fig. 22**) and can remain stable for months in the ambient storage environment.”

12. The authors should add more citations of recent works of other research groups focusing on high throughput synthesis of halide perovskites integrated with high throughput optical characterizations.

Thank you for the suggestion.

Action:

- We have now added a brief review on high-throughput characterization methods including accelerated measurement and rapid data analysis to the introduction section at Line 46-55: “In recent years, great effort has been made to develop high-throughput characterization capacities⁵, focusing on both faster data acquisition and data analysis (Supplementary Fig. 1). For example, time-evolved photoluminescence (PL) characterization and automated peak-fitting algorithms have been developed to study the broadband emission in mixed Sn-Pb 2D perovskite microcrystals synthesized by pipetting robots⁶. Additionally, batch-

processing algorithms⁷ and machine learning^{8,9} have been employed in data analysis pipelines, providing automated analysis of peaks in grazing-incidence wide-angle X-ray scattering (GIWAXS) and assisting identification of structural dimensions. However, a gap remains between the throughput of synthesis and characterization, posing a challenge for researchers to develop rapid characterization tools that can provide more accurate and comprehensive material properties.”

13. In supplementary figure 14, which data/figures is gradient printed, and which one is spin coated samples. please clarify that.

Action:

- We have clarified in the legend of **Supplementary Figure 21** at Line 376-377: “The grey dots represent bandgaps of spin-coated films. The colored dots represent average bandgaps of high-throughput samples from three rounds of experiments.”

14. The authors claim that they have generated 100,000 band gap data, no ML analysis has been done on the data. I realized they have done automatic bandgap extraction. What is the reason for not using ML. I agree that we do not necessarily always need to perform ML algorithms, but it would be great to provide some thoughts on this.

Thank you for your suggestion. We agree that our dataset is indeed large, with nearly 1 million bandgap data points collected using our high-throughput characterization pipeline. However, these data are concentrated along six lines (**Figure 1**), rather than being randomly distributed across the quasi-ternary chemical space. Unlike scattered data points that cover the space more uniformly, our line-based sampling results in dense data regions along specific sequences, leaving other areas sparse or entirely unsampled. Applying machine learning under such conditions would lead to poor generalization, especially in sparsely sampled regions far from these six lines where predictions would likely be inaccurate and still require separate experimental validation. Therefore, we did not apply machine learning in this work. Instead, the focus of our study is to demonstrate the capability of high-throughput optical characterization for exploring large chemical spaces.

Reviewer #2 (Remarks to the Author):

Sheng and Ji et al present a concise study outlining the importance of developing high-throughput (HT) analysis techniques to complement high-throughput experimental capabilities. This work uses perovskite-inspired materials to explore a wide range of compositions through HT workflows and subsequently analyzes their bandgaps through a hyperspectral imaging method connected with automated analysis. The primary focus of the research itself is on the optoelectronic properties and how variations in bandgap are related to different material compositions along with the effects of phase separation.

Overall I find the article is appropriately detailed, thorough and presents new insights for the range of compositions explored, specifically in terms of the effects of nanoscale phase-separation on bandgap. I believe in terms of content and execution it is appropriate for publication in Nature Communications pending some minor modifications and some additional discussion/foreword.

We appreciate the reviewer for the careful reading and positive feedback on our manuscript. We are grateful that the reviewer thinks this work provides insights in bandgap behaviors and phase separation phenomenon in perovskite-inspired materials.

1. Though I believe it could go beyond the scope of a communication, some additional commentary on other high throughput methods would be appropriate to add, particularly to highlight the relative speed, throughput, and presumably non-destructive nature of the presented technique versus alternative experiments.

Thank you for the suggestion.

Action:

- We have now added a brief review on other high-throughput characterization methods to the introduction section at Line 46-55: “In recent years, great effort has been made to develop high-throughput characterization capacities⁵, focusing on both faster data acquisition and data analysis (**Supplementary Fig.1** and **Supplementary Table.1**). For example, time-evolved photoluminescence (PL) characterization and automated peak-fitting algorithms have been developed to study the broadband emission in mixed Sn-Pb 2D perovskite microcrystals synthesized by pipetting robots⁶. Additionally, batch-processing algorithms⁷ and machine learning^{8,9} have been employed in data analysis pipelines, providing automated analysis of peaks in grazing-incidence wide-angle X-ray scattering (GIWAXS) and assisting identification of structural dimensions. However, a gap remains between the throughput of synthesis and characterization, posing a challenge for researchers to develop rapid characterization tools that can provide more accurate and comprehensive material properties.”

- We have also added a new supplementary figure and a supplementary table to exhibit the comparison of throughputs of different characterization methods, reproduced below:

Supplementary Figure 1. Throughput comparison of different characterization methods.

Supplementary Table 1. Throughput comparison of different characterization methods

Method	Acquisition Time/ Measurement	Damage to Sample	Accessibility	Ref
XRD	5-60 mins	moderate	Widely available	[5]
UV-Vis	1-5 mins	low	Widely available	empirical
SEM	2-10 mins	moderate	Widely available	empirical
High-throughput GIWAXS	10-60 s	moderate	Limited access	[7]
High-throughput PL	5-10 s	moderate	customized	[6]
High-throughput EDX	80-105 s	moderate	customized	[35]
High-throughput bandgap	10-20 μ s	low	customized	This work

[5] Miracle, D. B., Li, M., Zhang, Z., Mishra, R. & Flores, K. M. Emerging Capabilities for the High-Throughput Characterization of Structural Materials. *Annu. Rev. Mater. Res.* 51, 131–164 (2021).

[6] Foadian, E. et al. Decoding the Broadband Emission of 2D Pb-Sn Halide Perovskites through High-Throughput Exploration. *Adv. Funct. Mater.* 34, 2411164 (2024).

[7] Yang, J. et al. Accelerating materials discovery by high-throughput GIWAXS characterization of quasi-2D formamidinium metal halide perovskites. Preprint at <https://doi.org/10.26434/chemrxiv-2023-x7sfr-v3> (2023).

[35] Thienhaus, S., Naujoks, D., Pftzing-Micklich, J., König, D. & Ludwig, A. Rapid Identification of Areas of Interest in Thin Film Materials Libraries by Combining Electrical, Optical, X-ray Diffraction, and Mechanical High-Throughput Measurements: A Case Study for the System Ni–Al. *ACS Comb. Sci.* 16, 686–694 (2014).

2. In addition, there is a great deal of emphasis on the number of data points collected and in what timeframe. This would be well complemented by discussion about how many data points are necessary. This discussion is partially present in the SI through the experiments exploring batch to batch consistency but could be added in a concluding sentence in the discussion or through analysis of subsets of the data collected to identify the necessary minimum for the system. Further scaling and extension to infinite parameter spaces or replicates is not necessarily desirable or practical and I believe more explicit commentary on how much data needs to be collected to be meaningful would be worthwhile.

We agree with your point. It is unnecessary to collect nearly one million data points to reveal bandgap bowing phenomenon in this compositional space. The purpose of emphasizing the number of data points is to demonstrate the high throughput of our method. From the perspective of the approach, once the repeatability of high-throughput synthesis is established (**Supplementary Figs. 1 and 2**), only one round of experiments for each sequence is needed, rather than the six rounds conducted in this work. Thus, the number of data points could be reduced to one-sixth of what we actually collected. From the perspective of revealing bandgap behaviors in a complex material space, the minimum number of data points required depends heavily on the properties of the target material space. Theoretically, the more data points collected, the more convincing the conclusion will be. With higher spatial resolution (i.e., more bandgap data points within droplets), phase separation and elemental segregation phenomena can be investigated at a more microscopic scale in nonuniform droplets. With higher compositional resolution (i.e., more synthesized compositions within gradients), more detailed characteristics of bandgap changes can be captured, such as abnormal spikes or dips in a narrow compositional range. However, if the target material space tends to have uniform mixing of elements after crystallization and a simple linear bandgap change across the gradient, only one data point is needed for each droplet, and 3-5 compositions are sufficient to display the linear trend. Since we have little prior knowledge about the explored material space, we aimed to collect as many data points as possible. With the help of

the high-throughput platform, this process was greatly accelerated. However, we must admit that our method may experience diminishing returns in most material spaces.

3. By extension it would be insightful to have the authors more explicitly outline their selection criteria for pursuing specific samples with the slower techniques that are necessary for more complete characterization of a given composition.

We measured TA on spin-coated films along all six sequences. We chose spin-coated films because transient absorption (TA) measurements require thin, uniform samples to ensure optimal signal quality, which the drop-cast samples with larger thickness could not provide. Since both spin coating and TA measurements are relatively slow processes, we used a 10% compositional increment in each sequence for spin coating. As shown in **Supplementary Figure 21**, the optical properties of droplets and spin-coated films were nearly identical. Moreover, the bandgap trends obtained from TA were in excellent agreement with those derived from our high-throughput rapid method, validating its effectiveness. Therefore, we did not select specific compositions for TA measurements. Rather, we reduced the number of samples by lowering the compositional resolution for practical reasons. The overall compositional coverage remains unchanged.

4. Prior to publication I believe reference 26 must be updated. Presently it points to a pre-print published by several of the current authors that explores a different perovskite system using what I believe are the same droplet-based HTS system and the same hyperspectral autocharacterization system used in this work. The pre-print in question was published in Nature Communications earlier this year (Siemenn, A.E., Aissi, E., Sheng, F. et al. Using scalable computer vision to automate high-throughput semiconductor characterization. Nat Commun 15, 4654 (2024)). I believe this study extends that one and serves as an excellent follow-up demonstration of the system capabilities. As a follow-up I believe this study should highlight the pre-existing nature of the tools more prominently and should include some discussion of how the creation of datasets using pre-existing tools will increasingly become important for this type of research. Presently reference 26 is only featured in the methods and the SI at the end of the article. I believe it should be introduced at the beginning of the article in the introduction. The authors should also take the time to explain the novelty between this article and the current manuscript. What is the new contribution to knowledge that the current manuscript brings and how does that go beyond the article published earlier this year highlighting why they both deserve to be published in nature communications?

Thank you for the suggestion.

Action:

- We have now modified the following text in the introduction section at Line 66-73, mentioning our prior work and emphasizing the novelty of the current manuscript: “To

extract the bandgap, we build upon an automatic bandgap extraction algorithm originally developed in prior work¹⁷ (**Fig. 1**) and modify it to adapt to a more complex material space (**Supplementary Fig.2** and **Supplementary Table.2**). The data acquisition process follows the method used in prior work. Diffusive reflectance spectra are collected using a hyperspectral imaging setup, with a throughput of 80 samples every 5 minutes. For data analysis, we develop a spatially resolved optical analysis technique to extract approximately 250 local bandgaps (at 140×140 μm resolution) for a single droplet sample. This technique absorbs maximum information from the scan and creates a statistical distribution of bandgaps for a given material composition, allowing us to evaluate intra-sample homogeneity.”

- Besides, we have added a supplementary table demonstrating the main differences on algorithms between the current manuscript and prior work, reproduced below:

Supplementary Table 2. Differences in bandgap extraction algorithms

	Our work	Previous work¹⁷
Range	Full data range	Manual selection of target bandgap range
Resolution	Spatially-resolved bandgaps (N x N spectrum)	Average bandgap of each droplet (1 spectra)
Fitting	Linear regression on maximum difference	Linear regression on detected peaks

[17] Siemenn, A. E. et al. Using scalable computer vision to automate high-throughput semiconductor characterization. Nat. Commun. 15, 4654 (2024).

- Finally, we showed some failure modes of the previous code as well as the correct calculation of current codes in a newly added supplementary figure reproduced below:

Supplementary Figure 2. Comparison of bandgap extraction results between previous work and current manuscript.

Reviewer #3 (Remarks to the Author):

Reviewer #4 (Remarks to the Author):

Sheng, Ji et al. describe a high-throughput setup consisting of a rapid inkjet-printing-like setup complemented by diffuse reflectance and further augmented by transient absorption spectroscopy on perovskite-inspired Cs/Bi/Sb/I/Br mixes. Overall the study is interesting, but there remain several points that need clarification, before I can recommend publication.

Thank you for the thorough reading of the manuscript and the insightful questions and suggestions. We are grateful that the reviewer feels the study is interesting.

1. Visualization: While the color-maps are aesthetic, in some cases they make it hard to properly read the graphs. Prime examples are Figure S1 and S2 (white on a white background ...) Histograms would be much more meaningful (for Fig. S2, I'd even suggest correlation curves comparing the different rounds, much simpler)

Thank you for the suggestion. We agree that these figures are not straightforward.

Action:

- We have now added two supplementary figures. The first one shows the histograms of bandgap distributions of selected compositions in each round for the six sequences. The profiles provide a more straightforward understanding of the repeatability.

Supplementary Figure 6. Bandgap distribution histograms of 5 droplets with the same estimated compositions in three rounds of gradient printing for each sequence.

- Here, we added the histograms of bandgap distributions of five droplets evenly distributed in batch-printed samples, reproduced below:

Supplementary Figure 4. Bandgap distribution histograms of 5 droplets in 9 batch-printed samples. The number represents the index of droplets in the batch-printed samples.

- We have added extra description in Supplementary Note 2 at Line 73-75: For better visualization, we also plotted histograms of the bandgap distributions for the same compositions in the gradient for each round (Supplementary Figure 6).

2. The observation of a high degree of heterogeneity within a single droplet is an interesting observation. The authors should provide more information here. Is the (in-)homogeneity driven by a particular parameter? precursor concentration, pre-agitation, annealing time/temperature? For some select samples (e.g., most vs. least homogeneous) more in-depth characterization would be valuable to support the "general" validity of the HT results obtained.

Thank you for raising this point.

Action:

- We selected the composition $\text{Cs}_3(\text{Bi}_{0.5}\text{Sb}_{0.5})_2(\text{Br}_{0.5}\text{I}_{0.5})_9$ which has the largest degree of mixing and performed EDS mapping on a certain area of a drop-casted droplet. We added

a supplementary figure reproduced below. As shown in the figure, the distributions of B-site elements (Bi and Sb) and X-site elements (Br and I) are non-uniform in the selected area and have different patterns, supporting the heterogeneity of compositions in high-throughput synthesized samples.

Supplementary Figure 11. SEM-EDS mapping on drop-casted sample with the composition $\text{Cs}_3(\text{Bi}_{0.5}\text{Sb}_{0.5})_2(\text{Br}_{0.5}\text{I}_{0.5})_9$.

- We have now added in **Supplementary Note 3** at Line 116-118: To further support the XRD results, EDX mapping (**Supplementary Figure 11**) was performed on drop-casted samples with the composition $\text{Cs}_3(\text{Bi}_{0.5}\text{Sb}_{0.5})_2(\text{Br}_{0.5}\text{I}_{0.5})_9$.
- We have modified the main text at Line 135-137: However, we notice the printed droplets as well as the spin-coated films are not always spatially uniform due to crystallization kinetics and thermodynamic stability (**Supplementary Note. 6** and **Supplementary Fig. 11**).
- We have also added a **Supplementary Note 6**: Sample heterogeneity can be influenced by many factors. From a thermodynamic perspective, some mixed compositions are more stable when they decompose into multiple different compositions because the increasing entropy cannot compensate for the increasing enthalpy. Some compositions are inclined to be well-mixed since the mixing entropy enhances phase stability. From a kinetic perspective, the degree of heterogeneity is affected by crystallization conditions such as solvent, annealing temperature, and time. In this case, heterogeneity is primarily driven by the form factor of the droplets. During the drop-casting process, thick droplets require a longer time for solvent evaporation, leading to a slower crystallization process and a higher

degree of heterogeneity. This heterogeneity is unavoidable due to the synthesis method, even if we change other parameters like precursor concentration and annealing temperature.

3. The "mixed zone" bandgap regions observed are highly interesting. For many 3D ABX₃ perovskites phase separation is a dynamic process driven by light, heat, etc. Are the phases directly separated here? Considering the extremely fast characterization (at least the UV-Vis part) an observer-effect can be likely excluded...? E.g., by measuring the same sample before and after a short light ageing interesting information could be gained and also highlight another feature of the workflow, without me asking for this to be done for all samples, of course, but e.g. Fig. 2c provides a broad "mixed phase region".

We agree that many 3D perovskite phases can have light-induced phase separation. But in this manuscript, phases separation in $A_3B_2X_9$ perovskite-inspired materials occurred after crystallization instead of being driven by light illumination. Here, we compared the bandgap distributions of MA_{0.5}FA_{0.5}Pb(I_{0.5}Br_{0.5})₃ spin-coated film and Cs₃(Bi_{0.5}Sb_{0.5})₂(Br_{0.5}I_{0.5})₉ droplet before (**Supplementary Figs 22. a and c**) and after 10 minutes of light aging (**Supplementary Figs 22. b and d**) under the hyperspectral camera. As shown in the figure below, the droplet exhibits nonuniformity in bandgap distributions before light aging. Moreover, the patterns of bandgap distributions remain almost unchanged after 10 minutes of light illumination. In contrast, the MA_{0.5}FA_{0.5}Pb(I_{0.5}Br_{0.5})₃ spin-coated film initially shows a homogeneous bandgap distribution, with an obvious increase in heterogeneity in the bandgap distribution after light exposure. Therefore, the observer effect can be excluded due to the high stability of perovskite-inspired materials and the rapid characterization method.

Action:

- We have added clarification on stability of our samples in the results section at Line 87-88: "It shows no degradation after the optical measurement (**Supplementary Fig. 22**) and can remain stable for months in the ambient storage environment."
- We have added a supplementary figure reproduced which implies that no significant change happens before and after 10-minute illumination.

Supplementary Figure 22. Local bandgap maps of drop-casted $\text{Cs}_3(\text{Bi}_{0.5}\text{Sb}_{0.5})_2(\text{Br}_{0.5}\text{I}_{0.5})_9$ droplet and spin-coated $\text{MA}_{0.5}\text{FA}_{0.5}\text{Pb}(\text{I}_{0.5}\text{Br}_{0.5})_3$ film before and after 10-minute illumination. **a** and **b**, Bandgap maps of $\text{Cs}_3(\text{Bi}_{0.5}\text{Sb}_{0.5})_2(\text{Br}_{0.5}\text{I}_{0.5})_9$ droplet before and after 10-minute illumination under the light source of hyperspectral camera. **c** and **d**, Bandgap maps of $\text{MA}_{0.5}\text{FA}_{0.5}\text{Pb}(\text{I}_{0.5}\text{Br}_{0.5})_3$ film before and after 10-minute illumination under the light source of the hyperspectral camera.

4. For some of the select mixes with multiple bandgaps the actual spectra would be highly appreciated. As far as I know the work from the group, systems with multiple bandgaps haven't come up yet and hence I am a bit skeptical as to how the algorithm deals with that?

Thank you for pointing this out. The target composition in the precursor does not exhibit multiple bandgaps. However, after crystallization, the sample deviates from the target composition and becomes compositionally heterogeneous. Therefore, the multiple bandgaps result from the non-uniformity in the compositions on a microscopic scale. All the hypercubes with raw diffusive reflectance spectra and bandgap extraction results can be found at <https://github.com/shengf22/High-throughput-bandgap-mapping.git>.

Action:

- We have added clarifications in the results section at Line 170-172: “The observation of multiple optical bandgaps within certain compositional regions is highly suggestive of more than one semiconductor phase co-existing with a single printed droplet.”

5. Complementary XRD would be also very valuable to show the presence of two phases along the same lines of thinking.

Action:

- We have now added a supplementary figure with pXRD on drop-casted samples (**Supplementary Fig. 9a**) and XRD on spin-coated samples (**Supplementary Fig. 9b**) for sequence $\text{Cs}_3\text{Sb}_2\text{I}_9$ - $\text{Cs}_3\text{Bi}_2\text{Br}_9$ reproduced below:

Supplementary Figure 9. PXR and XRD results on sequence $(\text{Cs}_3\text{Sb}_2\text{I}_9)_{1-x}(\text{Cs}_3\text{Bi}_2\text{Br}_9)_x$. a, PXR patterns of drop-casted crystals. b, XRD patterns of spin-coated films.

- We have added more discussions in Supplementary Note 3: “As shown in **Supplementary Fig. 9a**, the pattern of the composition $(\text{Cs}_3\text{Sb}_2\text{I}_9)_{0.75}(\text{Cs}_3\text{Bi}_2\text{Br}_9)_{0.25}$, located in the 0D-2D structural transition region, implies the coexistence of 0D and 2D structures, with the peaks at 26.2° and 30.5° corresponding to the 0D structure and the peak at 27.8° corresponding to the 2D structure. The slight deviations from the reference patterns result from lattice changes after mixing B-site and X-site ions. For the composition $(\text{Cs}_3\text{Sb}_2\text{I}_9)_{0.5}(\text{Cs}_3\text{Bi}_2\text{Br}_9)_{0.5}$, the pattern demonstrates a 2D structure when compared to the reference pattern $\text{Cs}_3\text{Bi}_2\text{Br}_9$, with all peaks shifting to the left. Additionally, the peaks are broadened and asymmetric (also shown in **Supplementary Fig. 10**), indicating elemental segregation. However, it is challenging to identify the exact phases that share the same 2D structure but possess slightly different compositions. For the composition $(\text{Cs}_3\text{Sb}_2\text{I}_9)_{0.75}(\text{Cs}_3\text{Bi}_2\text{Br}_9)_{0.25}$, which

just exceeds the mixed-phase region, the main peaks become thinner compared to $(\text{Cs}_3\text{Sb}_2\text{I}_9)_{0.5}(\text{Cs}_3\text{Bi}_2\text{Br}_9)_{0.5}$. The results of thin-film XRD generally show broader peaks compared to pXRD due to lower crystallinity, making it difficult to detect elemental segregation by peak width. However, it is easier to identify 0D-2D structural changes with characteristic patterns. Clearly, $\text{Cs}_3\text{Bi}_2\text{Br}_9$, $(\text{Cs}_3\text{Sb}_2\text{I}_9)_{0.75}(\text{Cs}_3\text{Bi}_2\text{Br}_9)_{0.25}$, and $(\text{Cs}_3\text{Sb}_2\text{I}_9)_{0.5}(\text{Cs}_3\text{Bi}_2\text{Br}_9)_{0.5}$ exhibit similar diffraction patterns with a gradual left shift, indicating that all of them have a 2D $\text{Cs}_3\text{Bi}_2\text{Br}_9$ structure. However, $(\text{Cs}_3\text{Sb}_2\text{I}_9)_{0.75}(\text{Cs}_3\text{Bi}_2\text{Br}_9)_{0.25}$ shows a mixture of both 0D and 2D characteristics, aligning with the conclusion that the structural transition occurs in the range of $x = [0.10, 0.30]$ in the sequence $(\text{Cs}_3\text{Sb}_2\text{I}_9)_x(\text{Cs}_3\text{Bi}_2\text{Br}_9)_{1-x}$.”

[8] Sun, S. et al. Accelerated Development of Perovskite-Inspired Materials via High-Throughput Synthesis and Machine-Learning Diagnosis. *Joule* 3, 1437–1451 (2019).

6. I am not sure why the authors chose to perform TAS on the samples, as there is no real additional information derived from it in my opinion. The authors write: "We observed a strong correlation between the lowest-energy GSB and the steady-state optical bandgap from the hyperspectral measurements" and I "visually" agree. Why did the authors not simply correlate the two?

We appreciate the reviewer’s insightful comment. The primary reason for performing transient absorption spectroscopy (TAS) was to confirm that the material remains a direct bandgap semiconductor throughout the observed changes, thereby validating the correlation with the steady-state optical bandgap. If a direct-to-indirect transition or the presence of deep trap defect states were involved, such a correlation would not be valid.

As a result, TAS allows us to confirm the absence of deep trap states or a direct-to-indirect transition, as no significant changes in bandgap oscillator strength were observed. Another key motivation for conducting TAS was to extract carrier kinetics from the time-resolved transient absorption spectra, which provided insights into phase separation.

Action:

- Based on the reviewer’s suggestion, we have revised the phrasing from "strong correlation between" to "the steady-state optical bandgap from the hyperspectral measurements" to more accurately reflect our findings at Line 199-200: “We observed that the lowest-energy GSB is consistent with the steady-state optical bandgap from the hyperspectral measurements (Fig. 3a and b).”

7. The pump fluence of the TA measurement is very high (200 $\mu\text{J}/\text{cm}^2$, roughly 10,000x sun equivalent), so there is a high chance of measurement-induced artifacts, can the authors exclude that, e.g., by sequential measurements (I read the "The transient absorption spectroscopy was also

conducted in a high-throughput way. Each film was scanned only once and the average measuring time was around 10 min" paragraph, but I doubt it.

We appreciate the reviewer's concern regarding potential measurement-induced artifacts due to the high pump fluence used in the transient absorption (TA) measurements. To address this, we conducted additional tests on our samples by performing 2–3 sequential sweeps. As shown in the spectra and kinetic traces below, the data from the first, second, and third sweeps exhibit no significant variation, either in the absolute intensity of the spectral features or in the kinetic profiles. This confirms that the samples were not degraded by the laser exposure during the measurement process.

Action:

- We have added a supplementary figure reproduced as below:

Supplementary Figure 18. TA spectra of four pure compositions in three rounds of scan.

- We have modified the main texts in the results section at Line 239-242: Given the similar spectral characteristics across three rounds of sweeps and the absence of degradation during measurement (**Supplementary Fig. 18**), transient absorption spectroscopy was conducted in a high-throughput manner, with each film scanned only once and an average measuring time of approximately 10 minutes.

8. Can the authors correlate what they write to be correlated? e.g., GSB lifetime with "disorder" in the bowing region?

We appreciate the reviewer's feedback on the clarity of our correlation. We acknowledge that the degree of disorder cannot be directly extracted from the TAS measurements. To ensure a more precise interpretation, we have revised our description to focus solely on the observed transient spectral features and indicate whether the corresponding region aligns with the bowing position.

9. Another problem is that all samples are being treated the same here, but depending on the bandgap, bandgap-type (direct/indirect), quantum confinement, etc. things change. E.g., somewhat counterintuitively, at high fluences, for a direct-bandgap semiconductor without confinement (and it is not clear which of the compositions are which, so confined, direct or indirect bandgap?), one enters the radiative regime, which in turn leads to faster (compared to trap-assisted) recombination. Similarly confined vs. 3D systems behave vastly different in their ultrafast dynamics, so I am highly skeptical about their comparability in the way it is presented.

We appreciate the reviewer's insightful remarks regarding the different recombination mechanisms in semiconductors. In our study, the 3-2-9 phase perovskite-derived materials are fundamentally halide-based materials, with exciton wavefunctions constrained within only a few unit cells. This is in contrast to materials like CsPbI₃, where excitons tend to be highly delocalized and behave more like free carriers.

Given that the excitons in our system remain highly localized, we expect their recombination to predominantly occur in the excitonic regime, which is well described by mono-exponential decay. This allows us to meaningfully compare carrier lifetimes across our samples within this excitonic framework.

Reviewer #4 (Remarks on code availability):

not tested